# Characterizing Binding Interactions That Are Essential for Selective Transport through the Nuclear Pore Complex

**DOI:** 10.3390/ijms221910898

**Published:** 2021-10-08

**Authors:** Kathleen M. Lennon, Mohammad Soheilypour, Mohaddeseh Peyro, Devin L. Wakefield, Grace E. Choo, Mohammad R. K. Mofrad, Tijana Jovanovic-Talisman

**Affiliations:** 1Department of Molecular Medicine, Beckman Research Institute of the City of Hope Comprehensive Cancer Center, Duarte, CA 91010, USA; klennon@coh.org (K.M.L.); dwakefie@gmail.com (D.L.W.); gracechoo23@gmail.com (G.E.C.); 2Department of Bioengineering and Mechanical Engineering, Molecular Cell Biomechanics Laboratory, University of California, Berkeley, CA 94720, USA; soheilypour@berkeley.edu (M.S.); peyro@berkeley.edu (M.P.); 3Molecular Biophysics and Integrative Bioimaging Division, Lawrence Berkeley National Laboratory, Berkeley, CA 94720, USA

**Keywords:** nuclear pore complex, FG Nups, nuclear transport receptors, NPC barrier mimic, agent-based modeling, molecular dynamics, single molecule localization microscopy (SMLM)

## Abstract

Specific macromolecules are rapidly transported across the nuclear envelope via the nuclear pore complex (NPC). The selective transport process is facilitated when nuclear transport receptors (NTRs) weakly and transiently bind to intrinsically disordered constituents of the NPC, FG Nups. These two types of proteins help maintain the selective NPC barrier. To interrogate their binding interactions in vitro, we deployed an NPC barrier mimic. We created the stationary phase by covalently attaching fragments of a yeast FG Nup called Nsp1 to glass coverslips. We used a tunable mobile phase containing NTR, nuclear transport factor 2 (NTF2). In the stationary phase, three main factors affected binding: the number of FG repeats, the charge of fragments, and the fragment density. We also identified three main factors affecting binding in the mobile phase: the avidity of the NTF2 variant for Nsp1, the presence of nonspecific proteins, and the presence of additional NTRs. We used both experimentally determined binding parameters and molecular dynamics simulations of Nsp1FG fragments to create an agent-based model. The results suggest that NTF2 binding is negatively cooperative and dependent on the density of Nsp1FG molecules. Our results demonstrate the strengths of combining experimental and physical modeling approaches to study NPC-mediated transport.

## 1. Introduction

The nuclear pore complex (NPC) regulates selective and efficient transport of molecules across the nuclear envelope [1]. The NPC is an 8-fold symmetric macromolecular complex, which consists of multiple copies of approximately 30 proteins called nucleoporins (Nups) [2,3,4,5,6]. Nups have different functions within the NPC. Whereas transmembrane Nups anchor the NPC into the nuclear envelope, scaffold Nups create a passage through the nuclear envelope. A third type of Nups, called FG Nups, are disordered and contain repeated, hydrophobic phenylalanine and glycine (FG) motifs; they are grafted to the walls of the NPC and help create a selective barrier [4,7,8,9,10]. To achieve transport events on millisecond time scales [11,12,13,14], NPCs rely on many transient, weak interactions between FG Nups and nuclear transport receptors (NTRs) which can carry cargo molecules [15,16,17,18,19]. Both hydrophobic and electrostatic interactions between FG Nups and NTRs appear to be relevant for efficient transport [10,20,21,22].

The selective transport of macromolecules is essential for transcription, cellular signaling, and other fundamental cellular processes [1,23]. Unsurprisingly, errors in selective transport can lead to cellular dysregulation. Aberrant nucleocytoplasmic transport has been implicated in aging, neurodegenerative disorders (e.g., amyotrophic lateral sclerosis, Alzheimer’s disease, frontotemporal dementia, Huntington’s disease, Parkinson’s disease), viral infections, autoimmune disorders, and cancer [24,25,26,27]. In pathological conditions, Nups and/or NPC associated proteins can have missense mutations [28], protein alterations (e.g., truncation, fusion) [29,30], and anomalous expressions [31,32]. Unfortunately, efforts to fully understand the molecular mechanisms underlying NPC mediated transport in physiological and pathological conditions have been impeded by several factors: (1) the intrinsically disordered nature of FG Nups and their sequence heterogeneity; (2) the intricacy of the spatial geometry of the NPC; and (3) the complexity of the environment within the NPC. While important insight was obtained from studies in cellular models [11,12,13,14], much of our understanding about NPC mediated transport comes from in vitro and in silico (physics-based computational) studies assessing the roles of FG Nup assemblies, NTR binding to FG Nups, effects of NTRs on NPC morphology, and the role of cellular environment in effective transport [33,34,35,36,37]. Additionally, in vitro and in silico NPC (barrier) mimics have been powerful tools for interrogating transport, probing interactions between FG Nups and NTRs, and detailing NPC mediated transport [38,39,40,41,42,43,44,45,46,47,48,49,50,51,52].

One significant impediment to understanding the underlying mechanisms of NPC mediated transport has been the challenge of describing individual binding interactions between the FG domains and NTRs. This task is compounded by the cooperativity of the multivalent interactions between FG Nups and NTRs. However, the impact of multivalent interactions can become clearer when physical modeling approaches are used to interpret experimentally derived data [35,37,53]. Here, we combined a two-dimensional in vitro NPC barrier mimic with molecular dynamics and agent-based in silico models to investigate binding between a yeast FG Nup called Nsp1 and an NTR called nuclear transport factor 2 (NTF2). Nsp1 is an essential NPC protein [54]. The constituent FG and FSFG repeats of FG domain of Nsp1 are separated by hydrophilic linkers; this distribution of amino acids contributes to repulsive/cohesive molecular interactions of FG Nups and can facilitate an extended conformation for Nsp1 [20,48,55,56]. NTF2 is responsible for nuclear import of Ras related nuclear protein in its inactive, GDP bound, form (RanGDP) and helps maintain the directionality of nuclear transport [3,57]. NTF2 has two primary binding pockets associated with FG binding; the W7AI64A mutant leads to largely abolished binding to NPCs [58,59]. Together, these two proteins are an excellent model system to study the kinetics of NPC transport [14,48,58,60].

When designing our in vitro assay, we considered the following features of the native NPC: (1) in vivo, stationary FG Nups are closely packed within the NPC [10,61]; (2) nucleocytoplasmic transport is affected by mobile NTRs, cargo carrying NTRs (transport complexes), and nonspecific proteins (other cellular proteins within the NPC environment that do not bind to FG Nups) [8,33,36,38,48,62]. To account for these features, we generated a stationary phase of the NPC barrier mimic by covalently attaching a monolayer of yeast Nsp1FG fragments [48,63] to glass surfaces in an oriented manner [64]. Fluorescently labeled NTF2 was included in the mobile phase. Importantly, the mobile phase was a tunable environment which could include both nonspecific proteins and specific NTRs. Binding between Nsp1 and NTF2 was determined using fluorescence laser scanning. Furthermore, we used our experimental data and molecular dynamics modeling of Nsp1FG variants to guide our agent-based model (ABM). The combined approach allowed us to assess individual interactions between NTF2 and Nsp1FG, delineate factors important for efficient transport, and obtain new details on binding interactions between NTF2 and Nsp1.

## 2. Results

### 2.1. Characterizing the Stationary Phase of the NPC Barrier Mimic

Three His_6_-tagged Nsp1FG fragments with various numbers of FG repeats and different charges (Appendix A) were purified. SDS gels of purified proteins are shown in Appendix A. We used diazo chemistry to attach His_6_-tagged protein monolayers to coverslips in an oriented manner. His_6_-tagged polyethylene glycol (PEG) was used to saturate any remaining, unoccupied sites [64]. In our prior studies, we showed these activated surfaces preferentially bound His_6_-tagged proteins; assembled surfaces had saturated binding sites and were not perturbed by nonspecific proteins [64]. Additionally, His_6_-PEG coated surfaces (control experiments) show minimal background and exhibit non-fouling properties [64]. Importantly, the approach is compatible with quantitative single molecule localization microscopy (qSMLM) and allows for robust molecular counting [64]. Thus, we used qSMLM to determine the density of Nsp1FG fragments on the surfaces. First, each of the Nsp1FG fragments was fluorescently labeled with Alexa Flour 647 (AF647); on average, we covalently attached one dye molecule per protein (degree of labeling ~1). For each Nsp1FG fragment, a mixture of 1% labeled and 99% unlabeled protein was covalently attached onto activated glass coverslips [64] at two incubating concentrations: 3 μM and 30 μM. Subsequently, coverslips were imaged using qSMLM (Figure 1A). Detected molecular densities were quantified following a previously described method [64]. Low fluorescent protein content allowed us to detect sparse fluorophores and attain robust signal counting. Our results indicated there was no significant difference in detected surface densities between the Nsp1FG variants for the same incubating concentrations (Appendix A). When averaged for three variants and adjusted for fluorescent dilution, the obtained detected density of Nsp1FG was 689 Nsp1FG proteins/μm^2^ for layers incubated with 3 μM proteins and 1380 Nsp1FG proteins/μm^2^ for layers incubated with 30 μM proteins (Figure 1B). For corresponding detected densities, we simulated the grafting distances of the Nsp1FG proteins (Appendix A). For layers incubated with 3 μM proteins, the grafting distance was 13.1 ± 0.9 nm. For layers incubated with 30 μM proteins, grafting distance was 6.2 ± 0.2 nm, approaching that of the native NPC.

### 2.2. Molecular Dynamic Simulations to Determine Radii of Gyration

To assess the molecular conformation of our surfaces, we determined the radius of gyration for Nsp1FG5 and Nsp1FG12 fragments using a coarse-grained molecular dynamics model. In these simulations, the initial conformation of Nsp1FG variants were considered to be a straight line, perpendicular to the grafting surface. The obtained radii of gyrations are shown in Table 1 while trajectories of these simulations and snapshots are shown in the Appendix A. For simulated grafted polymer chains, the radii of gyration were 4.3 nm for Nsp1FG5 and 6.7 nm for Nsp1FG12. This represents the “mushroom” conformation of a single-grafted chain. The simulated grafting distance in the less densely grafted Nsp1FGs (3 μM incubating Nsp1FG concentration, Appendix A) was larger than the simulated radii of gyration, suggesting a “mushroom”-like conformation. On the other hand, the simulated grafting distance in the densely grafted Nsp1FGs (30 μM incubating Nsp1FG concentration, Appendix A) was closer than the dimension of an individual mushroom for Nsp1FG12, suggesting Nsp1FG12 and Nsp1FG18 appear to adopt the conformation more consistent with the “brush” regime.

### 2.3. Binding of NTF2-YFP onto Nsp1FG Monolayers

We assessed binding between Nsp1FG fragments and NTF2 in two grafting regimes. NTF2 tagged with yellow fluorescent protein (YFP) was purified with a two-step purification method using HisPur Cobalt chromatography and size exclusion chromatography (SDS gels of purified protein are shown in Appendix A). Either wild type (WT) or W7AI64A NTF2 mutant were used as NTRs in the mobile phase. Binding of NTF2-YFP to Nsp1FG fragment monolayers was assessed under several conditions. Both the composition of the mobile phase and grafting distance of the stationary phase were modulated. First, activated glass coverslips were treated with either His_6_-PEG (at 50 μM incubating concentration) or different Nsp1FG fragments (at 3 μM or 30 μM incubating concentrations followed by His_6_-PEG at 50 μM incubating concentration). Using a hydrophobic pen, glass slides coated with either Nsp1FG fragments or PEG (control) were divided into approximately 3 mm × 5 mm areas and placed into a 10 cm dish with damp kimwipes to prevent evaporation. Increasing concentrations of NTF2-YFP or NTF2W7AI64A-YFP (0–30 μM) were incubated onto either Nsp1FG or PEG areas for 45 min at room temperature. Three conditions were used in the mobile phase. (1) TBT buffer alone. (2) Nonspecific protein (10% bovine serum albumin (BSA)) in TBT buffer. This BSA concentration was chosen to mimic protein density within the cell [65]. (3) Nonspecific proteins and specific NTRs (10% Yeast Lysate (YL)) in TBT buffer. YL contains both nonspecific proteins, which do not bind to Nsp1, and various NTRs (specific proteins) which can compete with NTF2 for binding to Nsp1. The fluorescent intensity of YFP was detected using a Typhoon Imager (Figure 2).

Consistent with a number of previously reported in vitro assays that measured binding between FG Nups and NTRs [33], the apparent K_d_ values for NTF2-YFP in TBT binding to Nsp1FG surfaces for higher grafting density were in the nM range (Figure 3, black bars and Appendix A). These values are too tight for efficient NPC transport. Due to a lower number of FG domains, the apparent B_max_ value for NTF2-YFP binding to Nsp1FG5 was significantly lower than the apparent B_max_ values for NTF2-YFP binding to Nsp1FG12 and Nsp1FG18 (Figure 3, black bars and Appendix A). When 10% BSA was included in the soluble phase, the apparent K_d_ became much weaker for the binding of NTF2-YFP to all Nsp1FG fragments (μM range), closer to values expected for efficient nucleocytoplasmic transport [11,12,13,14]. Additionally, the apparent B_max_ values followed the same trend as with TBT buffer alone (lowest value for Nsp1FG5). For all Nsp1FG fragments, the apparent B_max_ values were significantly lower when 10% BSA was included (Figure 3, black bars vs. dark purple bars). This could be due to steric hinderance of some binding sites on Nsp1FGs at the physiologically relevant concentration of nonspecific proteins. Furthermore, the apparent K_d_ for NTF2W7AI64A-YFP (vs. WT NTF2-YFP, both in 10% BSA) decreased significantly for the Nsp1FG5 surface, increased slightly but significantly for Nsp1FG12 surface, and did not change for Nsp1FG18 surface. This effect could be due to the balance between the number of available FG domains, Nsp1FG fragment charge, and the reduced ability of the NTF2 double mutant to bind in the presence of nonspecific proteins. The apparent B_max_ for NTF2W7AI64A-YFP (vs. WT NTF2-YFP, both in 10% BSA) decreased significantly for all 3 surfaces; Nsp1FG5 surfaces had the lowest B_max_ value (Figure 3, dark purple bars vs. light purple bars).

We next probed the binding of WT NTF2-YFP to Nsp1FG fragments in the buffer containing 10% YL. Compared to 10% BSA condition (Figure 3, dark purple bars vs. dark orange bars), the apparent K_d_ did not change appreciably (except for slightly tighter binding on Nsp1FG12 surfaces). However, apparent B_max_ values were significantly reduced for all Nsp1FG fragments, likely due to binding of other available NTRs present in the YL. There was no detectable binding of NTF2W7AI64A-YFP to Nsp1FG fragments in buffer containing 10% YL (Figure 2A bottom right, Figure 3). NTRs present in YL are likely able to outcompete NTF2W7AI64A. Thus, our assay could detect the effect of nonspecific and specific proteins in the mobile phase. It was sensitive to both the composition of Nsp1FG fragments and avidity of the NTF2 variant.

We next examined the binding of NTF2-YFP to Nsp1FG monolayers that were grafted farther apart. Compared to WT NTF2-YFP in 10% BSA binding to densely grafted Nsp1FG (30 μM incubating concentration), the apparent K_d_ for WT NTF2-YFP in 10% BSA binding to less densely grafted Nsp1FGs (3 μM incubating concentration) decreased significantly for Nsp1FG5, increased slightly but significantly for Nsp1FG12 surfaces, and did not change for Nsp1FG18 surfaces. At the same time, the apparent B_max_ values were substantially reduced for Nsp1FG5 and Nsp1FG18. However, this decrease was much less pronounced for the Nsp1FG12 (Figure 3B, dark purple bars vs. dark teal bars). This result could be partly attributed to the effects of electrostatic interactions at this grafting regime: the charge at pH 7 is higher for Nsp1FG12 compared to Nsp1FG5 and Nsp1FG18 (Appendix A). Binding between NTF2W7AI64A-YFP (in 10% BSA) and less densely grafted Nsp1FGs (Figure 2B, top right) was low for Nsp1FG12, barely detectable for Nsp1FG5, and not detectable for Nsp1FG18 (apparent B_max_ had an appreciable value only for Nsp1FG12). Compared to WT NTF2-YFP in 10% BSA binding to less densely grafted Nsp1FGs, the apparent K_d_ values were moderately reduced for Nsp1FG12 (Figure 3, dark teal bars vs. light teal bars). Thus, the charge of Nsp1FG12 may contribute to interactions with mutant NTF2.

Finally, we probed the binding of NTF2 in buffer containing 10% YL to Nsp1FG monolayers grafted farther apart. Binding to Nsp1FG5 was not detected, likely due to the occupation of the few available binding sites by specific and nonspecific proteins at this grafting regime. Compared to the binding of NTF2-YFP in 10% BSA, the apparent K_d_ of NTF2-YFP in 10% YL did not change significantly and the apparent B_max_ decreased significantly for Nsp1FG12 (Figure 3, dark teal bars vs. dark yellow bars). Compared to the binding of NTF2-YFP in 10% BSA, the apparent K_d_ of NTF2-YFP in 10% YL increased significantly and the apparent B_max_ did not change appreciably for Nsp1FG18 (Figure 3, dark teal bars vs. dark yellow bars). We did not detect the binding between NTF2W7AI64A-YFP (in 10% YL) and Nsp1FG fragments (Figure 2B bottom right, Figure 3).

### 2.4. Agent-Based Modeling Suggests Multivalent NTF2-Nsp1FG Interactions

To further characterize the interactions between Nsp1FG fragments and NTF2, the in vitro NPC barrier mimic experiments were replicated using our ABM. Each ABM setup was first simulated with single binding assumption between Nsp1FG fragments and NTF2 (one NTF2 binds to one Nsp1). However, these simulations did not reproduce experimentally obtained apparent B_max_ values, which suggests that the single binding assumption was incorrect. This observation is consistent with previous reports of complex binding interactions between NTRs and FG Nups [47,48,49]. We next evaluated cooperative binding events, assigning different probabilities of binding for each individual event. We considered two cases: (1) a single NTF2 homodimer binds to multiple Nsp1FGs or (2) multiple NTF2 homodimers bind to a single Nsp1FG. For our experimental conditions (inclusion of nonspecific proteins), ABM results ruled out the first case (it would not provide sufficient binding events to achieve apparent K_d_ and B_max_ values from experiments) and supported the second case (K_d_ and B_max_ values that are in excellent agreement with experimental results, Table 2). ABM data also indicated that initial binding event(s) were much tighter compared to later binding event(s), indicating negatively cooperative binding of NTF2 to Nsp1FGs. On the one hand, simulations of Nsp1FG5 suggest two binding events at both grafting densities, but weaker binding on more densely grafted surfaces. On the other hand, simulations for Nsp1FG12 suggest three binding events with less densely grafted surfaces, but two binding events with more densely grafted surfaces. Thus, densely grafted Nsp1FG12 fragments are not fully accessible to NTF2, likely due to their close packing, consistent with the “brush” conformation.

## 3. Discussion

FG Nups are instrumental for maintaining rapid, bidirectional transport of macromolecules across the nuclear envelope and preventing translocation of nonspecific molecules. Previous work has established that both FG Nups and NTRs are important components of the selective barrier [34,38,46,47,48,49,66]. To determine the key factors that drive rapid, selective transport within the complex NPC environment, interactions between NTRs and FG Nups have been studied extensively [35,37]. Several studies have sought to elucidate these interactions through FG Nup assemblies onto beads [33,67], nanopores [38,40], or 2D planar surfaces [46,47,48,50,52]. While experiments with stationary FG Nups need to be interpreted carefully [51], they offer important insights into binding events. Here, we used a fluorescence assay to assess the macroscopic binding of NTF2 to 2D planar FG Nup surfaces. Since the experimental setup provides apparent K_d_ values and assessment of apparent binding capacity (B_max_), we complemented experiments with molecular dynamics simulation and ABM. The approach provided additional insight into the binding between Nsp1FG and NTF2.

Our experimental setup employed diazo chemistry on glass surfaces [64] to covalently attach His_6_-tagged FG Nup fragments in an oriented manner. We used FG Nup fragments with different numbers of FG repeats and charge (Appendix A). Because of its non-fouling properties and minimal background in fluorescence assays, His_6_-PEG was included to saturate unreacted sites. This assay is compatible with quantitative SMLM and allowed us to determine detected surface densities of Nsp1FG fragments for two incubating concentrations: 3 μM and 30 μM proteins (Figure 1). Using simulations, we next determined grafting distances for assembled protein monolayers (Appendix A) and complemented these data with radii of gyration obtained with molecular dynamics simulations in GROMACS (Table 1). Our results indicate that for 3 μM Nsp1FG incubating concentrations, we achieved a “mushroom” regime. In contrast, for 30 μM Nsp1FG incubating concentrations, grafting distances approached those in the native NPC and data was more consistent with the “brush” regime. This is in line with previous data on the average layer height of Nsp1FG fragments [48] and our ABM results. Our assembled surfaces allowed us to investigate binding between labeled NTRs and FG Nups using fluorescence scanning.

The binding constants between NTRs and FG Nups reported in in vitro studies are frequently too tight to account for the rapid transport seen in vivo [11,12,13,14]. In agreement with published reports [33], we found equilibrium binding of NTF2-YFP to Nsp1FGs to be in the nM range in TBT buffer (Figure 2 and Figure 3). In the presence of physiological concentrations of nonspecific proteins (10% BSA), binding was in the μM range. Several important conclusions stand out. (1) Higher apparent binding capacity was observed on more densely grafted surfaces; this is consistent with a higher number of available Nsp1FG molecules. (2) Increased grafting distance led to significant changes in the apparent K_d_ for shorter Nsp1FG variants; however, the apparent K_d_ was not significantly affected for Nsp1FG18. This effect could be due to the ratio between grafting distance and radius of gyration for different fragments and/or the larger number of available binding sites on Nsp1FG18. (3) When considering binding of NTF2 to Nsp1FGs, both the number of FG repeats and fragment charge play an important role. In future studies, the details associated with charge effects could be investigated using mutations on Nsp1FG fragments. (4) WT NTF2-YFP binds with higher apparent binding capacity to Nsp1FGs compared to mutant NTF2W7AI64A-YFP. On less densely grafted surfaces, NTF2W7AI64A-YFP binds appreciably only to Nsp1FG12. In this grafting density regime and in the presence of nonspecific proteins, charge interactions appear to dominate when hydrophobic interactions are reduced.

We investigated the effect of both specific and nonspecific proteins present in the mobile phase using YL. For NTF2-YFP, no significant difference in binding was observed for Nsp1FG18 surfaces at either grafting density; slightly weaker binding was observed on Nsp1FG12 surfaces for the lower grafting density; and there was no binding to Nsp1FG5 surfaces for the lower grafting density. For NTF2W7AI64A-YFP no binding to Nsp1FG fragments was detected in all cases. NTRs from YL appear to fully displace mutant NTF2. This is consistent with previous reports in cells [59]. Altogether, our experimental data suggest that both the stationary phase and mobile phase play important roles in creating an effective selective barrier.

To further investigate the interaction between Nsp1FG fragments and NTF2 molecules, we developed an ABM of our experiments (mobile phase: WT NTF2-YFP with 10% BSA in TBT). ABM results suggest that multiple NTF2 homodimers could bind to a single Nsp1FG. However, it should be noted that in native NPCs, complex binding scenarios could be envisioned: (1) a single NTF2 homodimer could bind to multiple Nsp1FGs or multiple FG domains on a single Nsp1FG; (2) multiple NTF2 homodimers could bind to a single Nsp1FG or to multiple Nsp1FGs; (3) a combination of interaction could occur. While our ABM simulations did not directly test these cases, we speculate that, in native NPCs, when many NTRs are present, NTF2 dimers could bind to a pool of accessible (and entropically affordable) FG binding sites. The initial binding events had lower K_d_ values and subsequent events had higher K_d_ values, which is consistent with reported negative cooperativity of NTR binding to FG Nups [47,48,49]. For example, after the first and/or second binding events, Nsp1FG binding regions could be less accessible to NTF2 dimers due to molecular crowding or structural rearrangements. In the complex environment of the NPC, prebound NTRs likely modulate binding of incoming NTRs. Altogether, by combining experimental and physical modeling approaches, we identified important factors that affect binding between NTRs and FG Nups and provided insight on binding interactions and binding stoichiometry.

## 4. Materials and Methods

**Materials.** PEG-His_6_ was synthesized using Standard solid-phase N-αFmoc chemistry on a CS136XT peptide synthesizer (C S BIO, Menlo Park, CA) at the City of Hope Peptide Synthesis Core as previously described [64].

**Proteins.** His-tagged Nsp1FG5 (residues 262–359) and Nsp1FG12 (residues 262–492) in pET30ATEV, and Nsp1FG18 (residues 262–603) in pET30A plasmids were kindly provided by Dr. M. Stewart [58]. Nsp1FG variants were transformed into BL21DE3 cells (New England Biolabs, Ipswich, MA, USA). Cells grown in LB medium with kanamycin selection were induced with 1mM IPTG at optical density (OD) 0.7 and harvested after 16 h at 25 °C. Cells were resuspended in buffer A (50 mM sodium phosphate pH 8.0, 300 mM NaCl, 50 μM EDTA, 5 mg/mL 6-Aminohexanoic acid (Alfa Aesar, Ward Hill, MA, USA), protease inhibitor cocktail tablet (Sigma, St. Louis, MO, USA: AEBSF 2 mM, Phosphoramidon 1 mM, Bestatin 130 mM, E-64 14 mM, Leupeptin 1 mM, Aprotinin 0.2 mM, Pepstatin A 10 mM), and lysed with a French press. Cell debris was removed with a 45-min spin at 45,000 RPM in a Ti-70 rotor (Beckman, Brea, CA, USA). Nsp1FG variants were purified using HisPur Cobalt resin (Thermo Scientific, Waltham, MA, USA) and buffer was exchanged to PBS.

For measurements of surface density, Nsp1FG variants were labeled with Alexa Fluor 647 NHS ester (Invitrogen, Waltham, MA, USA). A volume of 100 μL of 1 mg/mL protein in PBS supplemented with 2 mM of sodium bicarbonate containing buffer was incubated with 3 times molar excess of dye (30 min at room temperature). After quenching with 150 mM hydroxylamine HCl pH 8.5, unbound dye molecules were removed using Micro bio-spin 6 gel filtration columns (Bio-Rad, Hercules, CA, USA) equilibrated with PBS, and any remaining aggregates were removed with Nanosep 300kD filers (PALL, Port Washington, NY, USA). The degree of labeling for Nsp1FG-AF647 was determined using a Nanodrop; the values ranged between 0.8 and 1.2 dye per protein.

His-tagged NTF2-YFP and NTF2W7AI64A-YFP in pET21b [38] were transformed into BL21DE3 cells (New England Biolabs, Ipswich, MA, USA). Cells grown in LB medium with ampicillin selection were induced with 0.1 mM IPTG at OD 0.65 and harvested after 16 h at 25 °C. Cell were resuspended in buffer B (50 mM sodium phosphate pH 7.5, 300 NaCl, 0.5% Tween-20, protease inhibitor cocktail tablet (Sigma, St. Louis, MO, USA), and lysed with a French Press. Cell debris was removed with a 45 min spin at 45,000 RPM in a Ti-70 rotor (Beckman, Brea, CA, USA). His-tagged NTF2 variants were purified in two steps using HisPur Cobalt Superflow resin (Thermo Scientific, Waltham, MA, USA) and a Superose 6 Increase 10/300 or Superdex 200 Increase 10/300 column (GE Healthcare, Chicago, IL, USA) equilibrated with TBT (20 mM HEPES pH 7.4, 110 mM potassium acetate, 2 mM MgCl_2_, 10 μM ZnCl_2_, 10 μM CaCl_2_, 0.1% Tween-20).

**Preparation of coverslips for measurement of Nsp1FG surface density.** 25-mm #1.5 coverslips (Warner Instruments, Hamden, CT, USA) were cleaned [68], flame dried, and stored in a dry place away from light. Coverslips were activated as described previously [64]. Briefly, cleaned coverslips were incubated with concentrated (12N) HCl for 3 min, followed by several rinses with distilled water and absolute ethanol. Then coverslips were incubated with 9.4 mM ATMS (Gelest, Morrisville, PA, USA) in absolute ethanol for 30 min at room temperature, rinsed with absolute ethanol, air dried, and incubated with freshly prepared 260 mM HCl and 5.2 mM NaNO_2_, in distilled water, for 30 min at 4 °C. Coverslips were washed 3 times with ice cold sodium acetate, twice with water, and twice with PBS. A volume of 150 μL of either 3 μM or 30 μM Nsp1FG fragments (1% Nsp1FG-AF647 and 99% unlabeled Nsp1FG in PBS) were placed on the top of activated coverslips. After a 30 min incubation at room temperature, coverslips were washed twice with PBS followed by incubation with 50 μM PEG-His_6_ in PBS for 30 min at room temperature to fill unreacted sites. Surfaces were imaged immediately after preparation in Attofluor cell chambers (Life Technologies, Carlsbad, CA, USA) in 50 mM Tris (pH 8.0), 10 mM NaCl, and 10% glucose imaging buffer containing mercaptoethylamine (MEA, 100 mM) and GLOX (10% v/v).

**dSTORM imaging.** Measurements were performed on a 3D N-STORM super-resolution microscope (Nikon, Melville, NY, USA). The N-STORM system is a fully automatic Ti-E inverted microscope with a piezo stage on a vibration isolation table. This system includes a 100× 1.49 NA TIRF objective (Apo), N-STORM lens, λ/4 plate, and Quad cube C-NSTORM (97355 Chroma). The microscope has a Perfect Focus Motor, to maintain imaging at the appropriate focal plane. An MLC-MBP-ND laser launch included 405, 488, 561, and 647 nm lasers (Agilent, Santa Clara, CA, USA). Images were captured with an EM-CCD camera iXon DU897-Ultra (Andor Technology, South Windsor, CT, USA). Data was acquired using NIS-Elements 4.3 Software (Nikon, Melville, NY, USA). dSTORM images of 41 × 41 μm were collected with an exposure time of 10 ms. 10,000 frames were acquired for each field of view. For imaging AF647, the 647 nm laser power was ~120 mW. Fluorophore localizations (above 700 photons) were extracted from raw image data using NIS-Elements and drift correction was performed. The number of localizations were analyzed to determine the average number of fluorophore appearances (localizations) per molecule and average surface density using a custom MATLAB code [64].

**Charge for Nsp1FGfragments**. Nsp1FG variants were assessed for charge at pH 7.0 given the amino acid sequence using the Protein Calculator v3.4 (http://protcalc.sourceforge.net/, accessed on 14 August 2021) through C. Putnam, The Scripps Research Institute, USA. The pKa values used for the individual amino acids in these calculations are provided from Stryer Biochemistry, 3rd edition.

**NPC barrier mimic assay.** Individual His-tagged Nsp1FG variants were attached to standard microscopy slides (AmScope, 7101, Irvine, CA, USA) using ATMS/diazo chemistry as described above. Briefly, cleaned slides were activated with 1 mL of 9.4 mM ATMS, rinsed, air dried, and incubated with 1 mL of freshly prepared NaNO_2_. Slides were washed 3 times with ice cold sodium acetate, twice with water, and twice with PBS. After washing, the backs of the slides were dried with kimwipes and excess liquid was tapped off. Slides were incubated with 1 mL of Nsp1FG variants diluted in PBS (either 3 μM or 30 μM final concentration) or 1 mL of PBS alone (for PEG-His_6_ controls). After 30 min, slides were washed twice with PBS followed by covalent attachment of 50 μM PEG-His_6_ in PBS. Slides were rinsed 3 times with PBS and excess liquid was tapped onto a kimwipe. Slides were then divided into roughly 3 mm by 5 mm rectangles (a 4 by 8 grid) using an ImmEdge Hydrophobic Barrier PAP pen (Vector Labs, Burlingame, CA, USA).

Increasing concentrations of NTF2-YFP or NTF2W7AI64A-YFP in TBT were prepared and 3 μL was added to each rectangle. Where indicated, the mobile phase was supplemented with either 10% (w/v) filter sterilized BSA (Sigma, St. Louis, MO, USA) or 10% YL (w/v). YL was prepared from the BCY123 yeast strain kindly provided, as clarified lysate, by Dr. G. Pineda. Slides were incubated in humidified chambers for 30 min at 37 °C, rinsed with TBT 3 times, and bound YFP labeled protein was detected on a Typhoon Laser Imager (Amersham Biosciences, Amersham, United Kingdom). To detect YFP signal on glass coverslips, fluorescence acquisition mode was used with the 526 SP emission filter, 400 PMT with the Green Laser (532), and normal sensitivity. Images were collected with a 10-micron pixel size. Following imaging, custom code in MATLAB was used to select regions of interest (ROI) for each condition, within each 3 mm by 5 mm rectangle, and average intensity values were extracted. Average intensity values for each condition were fit using a Langmuir isotherm in Prism to produce the binding curves shown in Figure 2 and assess the binding of NTF2-YFP or NTF2W7AI64A-YFP to Nsp1FG variants. A two-site Langmuir isotherm was tested. However, this did not provide a good fit to the data collected in this experimental setup.

**Grafting distance simulations.** Using experimentally derived protein densities, synthetic localization data was prepared in MATLAB to estimate Nsp1FG grafting distances. Individual localizations representing Nsp1FG were randomly distributed (MATLAB *rand* function) within a 1 μm^2^ area. The total number of localizations were varied using the MATLAB *normrnd* function to reproduce average densities of 689 and 1379 proteins/μm^2^, corresponding to results from Nsp1FG surfaces incubated with 3 μM and 30 μM Nsp1FGs, respectively (Figure 1). The associated standard error for each concentration was also used as input for the MATLAB *normrnd* function to provide more realistic variation in the simulated densities. To prepare simulated localization data for FG Nups in the native NPC, an average density of 14,000 proteins/μm^2^ was used, along with a random number, generated from an inverse Gaussian distribution, to supply a small amount of variation in the average number of localizations. This distribution was prepared by collecting the frequency of spatial differences between FG Nups, as reported in Tagliazucchi et al. [22], and applying a fit to the data. With localization data in place, inter-point distances were then calculated using a modified version of the MATLAB function *pdist2*. This approach accommodated the standard metric for taking the Euclidean distance for each localization and its nearest neighbor, while subdividing the data into manageable batches to assist with computation and matrix size limits. Additional checks were provided to exclude duplicate overlapping localizations and ensure the smallest pairwise distance was collected. This process of randomly placing localizations within a defined area and calculating pairwise distances was iterated 15 times. Final simulated grafting distance averages are reported in Appendix A.

**Coarse-grained molecular dynamics simulations.** One-bead-per-amino-acid coarse-grained molecular dynamics was used to simulate the behavior of Nsp1 molecules. This coarse-grained model is specifically designed to explore the behavior of intrinsically disordered proteins. Amino acids are modeled as spherical beads with a mass of 120 Da and distance of 0.38 nm. The force-field of this model, was developed by Ghavami et al. [69]. This force-field takes into account bending and torsion potentials between neighboring beads, implicit solvent, ion screening effect, and hydrophobic and electrostatic interactions [10,69]. Langevin dynamics simulations were done using GROMACS molecular dynamics simulation software [70]. In the simulations, the system was minimized first, equilibrated for 50 ns and then run for 1μs with a time-step of 0.02 ps. Visualizations were done using VMD 1.9.3 [71]. Radius of gyration of the Nsp1 molecules was measured using “gmx gyrate” function in GROMACS.

**Agent-based modeling simulations.** ABM is a bottom-up complex systems approach for simulating the interactions between multiple independent entities, termed “agents”. The objective is to assess individual effect of diffusing and reacting agents on the overall system and predicting subsequent emergent phenomena [43,72]. The in vitro experiments were replicated using our ABM platform that was previously used for various applications regarding nucleocytoplasmic transport [73,74,75]. In our ABM, each agent, representing a single molecule or a homodimer, is characterized by its location and volume and is assigned probabilities representing its diffusion as well as potential binding/unbinding events [43,74]. A 3D environment of ~1 μm × 1 μm × 100 nm was simulated for each setup. Nsp1FG agents were immobilized and attached to the surface. NTF2 agents were free to move. Average grafting distance was used to randomly distribute Nsp1FG agents on the surface. NTF2 agents were added to the system with varied concentrations of 1, 2, 3, 5, 10, 30 μM. Size of Nsp1FG fragments was calculated through coarse-grained molecular dynamics simulations. Each simulation was run for three million steps to ensure that the system has reached equilibrium and the last 100 steps were used to count bound and free NTF2 molecules. Experimentally-derived K_d_ was used as the input to determine binding and unbinding probabilities between Nsp1FG fragments and NTF2. We have previously formulated how binding and unbinding constants (k_on_ and k_off_) could be converted into ABM probabilities (P_on_ and P_off_) [43]. We used a probability of 0.005 as P_on_ and, considering that K_d_ = k_off_/k_on_, calculated the respective P_off_. A sensitivity analysis was conducted, demonstrating that change in the initial assumed value for P_on_ does not significantly affect the overall outcome and that the P_off_/P_on_ ratio is the primary factor determining the overall apparent K_d_ and B_max_. Number of possible binding events and binding strength was varied in a systematic trial and error process until the obtained overall K_d_ and B_max_ from simulations are representative of the values obtained from experiments. Woolf-Hanes plot was used to calculate overall K_d_ and B_max_. Simulations of the final simulated setups were repeated three times and average and standard deviation values were reported.

## Figures and Tables

**Figure 1 ijms-22-10898-f001:**
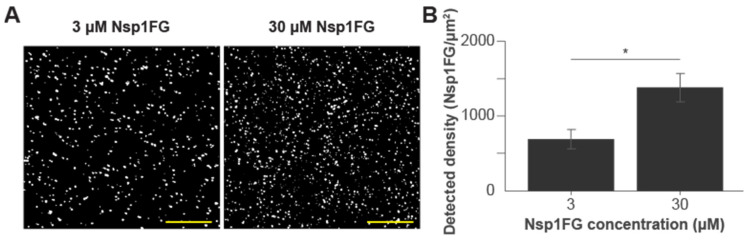
Average detected surface density of Nsp1FGs. Activated glass coverslips were incubated (at 3 μM or 30 μM) with one of three variants of Nsp1FG. (**A**) Representative ROIs showing localizations of Nsp1FG12-AF647. Scale bar, 5 μm. (**B**) The average detected density for Nsp1FG surfaces at 3 μM incubating concentration (*n* = 30 ROIs, 6 independent measurements for Nsp1FG5, Nsp1FG12, and Nsp1FG18) and at 30 μM incubating concentration (*n* = 30 ROIs, 6 independent measurements for Nsp1FG12 and Nsp1FG18); * *p* ≤ 0.02. Error bars represent SEM.

**Figure 2 ijms-22-10898-f002:**
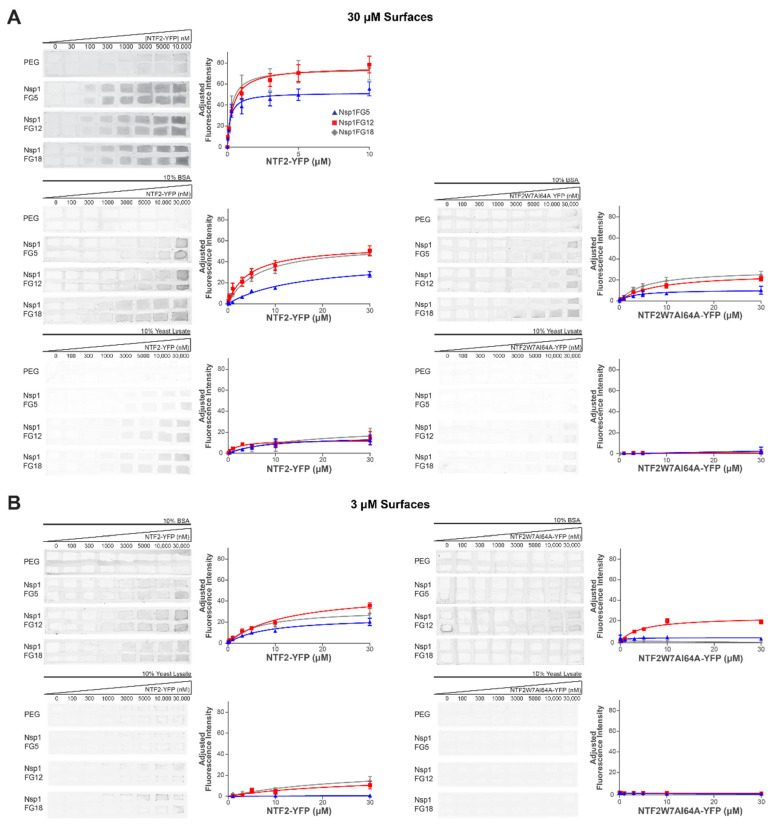
NTF2-YFP and NTF2W7AI64A-YFP binding on Nsp1FG and PEG surfaces. (**A**) Left: Representative fluorescent images and binding curves of NTF2-YFP; incubating [Nsp1FG variants] = 30 μM. Right: representative fluorescent images and binding curves of NTF2W7AI64A-YFP; incubating [Nsp1FG variants] = 30 μM. Error bars represent SEM. *n* = 8 ROIs, 4 independent measurements for TBT buffer and 10% BSA in TBT buffer; *n* = 4 ROIs, 2 independent measurements for 10% YL in TBT buffer. (**B**) Left: Representative fluorescent images and binding curves of NTF2-YFP; incubating [Nsp1FG variants] = 3 μM. Right: representative fluorescent images and binding curves of NTF2W7AI64A-YFP; incubating [Nsp1FG variants] = 3 μM. Error bars represent SEM. In each case, *n* = 6 ROIs, 3 independent measurements.

**Figure 3 ijms-22-10898-f003:**
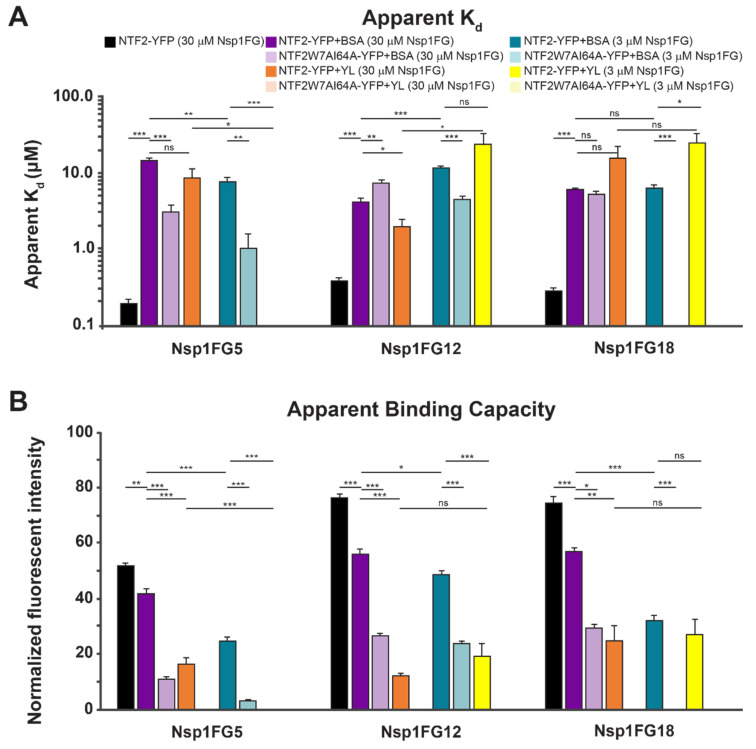
NTF2-YFP apparent K_d_ and binding capacity on Nsp1FG surfaces. (**A**) The apparent K_d_ of NTF2-YFP and NTF2W7AI64A-YFP for Nsp1FG surfaces. (**B**) The apparent binding capacity of NTF2-YFP and NTF2W7AI64A-YFP for Nsp1FG surfaces. In all cases, error bars represent SEM. ns = not significant (*p*-value > 0.05), * = *p*-value 0.05–0.001, ** = *p*-value 0.001–0.00001, *** = *p*-value < 0.00001. *p*-values are detailed in Appendix A.

**Table 1 ijms-22-10898-t001:** Radii of gyration of Nsp1FG variants as determined from our molecular dynamics model.

Nsp1FG Variant	Radius of Gyration (nm)
Nsp1FG5	4.3
Nsp1FG12	6.7

**Table 2 ijms-22-10898-t002:** Obtained values for binding between NTF2-YFP and Nsp1FG fragments from the ABM simulations.

	Input K_d1_	Input K_d2_	Input K_d3_	Apparent K_d_	Apparent B_max_
Nsp1FG5 + BSA 3 μM surfaces	5.4	16.2	--	8.16 ± 0.26	24.15 ± 0.19
Nsp1FG5 + BSA 30 μM surfaces	10.2	38.25	--	15.16 ± 1.03	45.63 ± 1.90
Nsp1FG12 + BSA 3 μM surfaces	5.45	5.45	40	11.87 ± 0.35	50.59 ± 2.66
Nsp1FG12 + BSA 30 μM surfaces	2.2	4.89	--	4.35 ± 0.24	58.23 ± 3.81

All K_d_ values are μM and errors are SEM.

## Data Availability

Not applicable.

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
