# Peer review of "Characterizing Binding Interactions That Are Essential for Selective Transport through the Nuclear Pore Complex"

_ijms, 2021, doi:10.3390/ijms221910898_

Round 1

Reviewer 1 Report

The topic is interesting and the manuscript is prepared very well. There are certain issues that need to be addressed before future steps:

  1. Please use full description first and then abbreviation. For instance, Nups, NTF2, and BSA in Abstract.
  2. Please remove horizontal lines from all Tables.
  3. Use the capital letters as same as figures in the figure legends.
  4. Newer references need to use. most of them are behind 2018.
  5. It's better to present some information about the role of NCT defects in different diseases. Authors can read the following paper as an example:
    • https://www.mdpi.com/1422-0067/22/8/4165
  6. The resolution of the figures is not good.

Author Response

The topic is interesting and the manuscript is prepared very well. There are certain issues that need to be addressed before future steps:

We thank the Reviewers for their helpful comments. As suggested, we provide more details about what possible roles the NPC plays in disease. We have included this topic (and additional references) in the Introduction Section.

1. Please use full description first and then abbreviation. For instance, Nups, NTF2, and BSA in Abstract.

We agree and have included full descriptions before the abbreviations.

2. Please remove horizontal lines from all Tables.

Horizontal lines have been removed from all Tables.

3. Use the capital letters as same as figures in the figure legends.

This has been corrected.

4. Newer references need to use. most of them are behind 2018.

We thank the Reviewer for this important point. We have extended the Introduction to include new references; some have been published recently.

5. It's better to present some information about the role of NCT defects in different diseases. Authors can read the following paper as an example:

    • https://www.mdpi.com/1422-0067/22/8/4165

We thank the Reviewer for this helpful suggestion. We have included a brief discussion about what role NPC-mediated transport may play in pathological states. We have included the suggested reference as an important example.

6. The resolution of the figures is not good.

We agree. The revised Figures are higher resolution.

Reviewer 2 Report

In the manuscript entitled "Characterizing binding interactions that are essential for selective transport though the nuclear pore complex", written by Lennon KM, Soheilypour M, Peyro M, Wakefield DL, Choo GE, Mofrad MRK and Jovanovic-Talisman T,  an in vitro method for investigating binding of nuclear receptor protein to nuclear pore complex protein is presented. The authors attached different variants of nuclear pore protein with FG chains to glass coverslip and analyzed binding of wt and mutated NTF-2, a receptor protein, labelled with fluorescent protein, to it. NTF2 binding affinity and binding capacity was analyzed, in dependence on the length of FG sequence, presence of mutation in NTF2, density of pore proteins on the glass etc. Agent based model simulations were done and results aim to give an insight into binding interactions regulating transport through nuclear pore.

The manuscript is well written. The Introduction gives an overview of the topic, results are clearly presented and methods well described. However, there are some questions. Three lengths of FG chains were used, FG5, FG12 and FG18. FG12 has different charge from FG5 and FG18. Both, length and charge could influence the binding. Is it possible to do experiments with proteins of different length with similar charge and with the same length and different charge? Another comment is dealing with Figure 2 and different combinations of wild type and mutant receptor protein binding to surface. Why were not all combinations done (i. e. there are no data on binding of mutant protein alone and combinations with 30 µM surface)?

Minor comments:

line 73: helps maintaining

line 113: Figure S1C

line 253: delete the sentence: All values...

line 414: slides

line 469: better explanation

units should be written separately from the numbers

Author Response

In the manuscript entitled "Characterizing binding interactions that are essential for selective transport though the nuclear pore complex", written by Lennon KM, Soheilypour M, Peyro M, Wakefield DL, Choo GE, Mofrad MRK and Jovanovic-Talisman T,  an in vitro method for investigating binding of nuclear receptor protein to nuclear pore complex protein is presented. The authors attached different variants of nuclear pore protein with FG chains to glass coverslip and analyzed binding of wt and mutated NTF-2, a receptor protein, labelled with fluorescent protein, to it. NTF2 binding affinity and binding capacity was analyzed, in dependence on the length of FG sequence, presence of mutation in NTF2, density of pore proteins on the glass etc. Agent based model simulations were done and results aim to give an insight into binding interactions regulating transport through nuclear pore.

The manuscript is well written. The Introduction gives an overview of the topic, results are clearly presented and methods well described. However, there are some questions. Three lengths of FG chains were used, FG5, FG12 and FG18. FG12 has different charge from FG5 and FG18. Both, length and charge could influence the binding. Is it possible to do experiments with proteins of different length with similar charge and with the same length and different charge? Another comment is dealing with Figure 2 and different combinations of wild type and mutant receptor protein binding to surface. Why were not all combinations done (i. e. there are no data on binding of mutant protein alone and combinations with 30 µM surface)?

We thank the Reviewer for raising the important issue about assessing additional conditions with more densely grafted surfaces. We have now performed additional experiments with WT and mutant NTF2 in the presence of yeast lysate. Experiments were performed for all 3 Nsp1FG fragments (FG5, FG12 and FG18) incubated at 30 µM concentration. Our results are shown in Figures 2 and 3; they are presented in the Results section and discussed in the Discussion section. We agree that these suggested experiments are helpful for demonstrating two important considerations when investigating binding of NTRs to FG Nups: both the charge and number of hydrophobic repeats on FG Nup fragments appear to affect binding. As pointed by the Reviewer, we now discuss how the importance of charge vs. hydrophobicity could be demonstrated in future experiments using mutations on Nsp1 fragments,

Minor comments:

All minor comments are now addressed, thank you for pointing out these issues.

line 73: helps maintaining

line 113: Figure S1C

line 253: delete the sentence: All values...

line 414: slides

line 469: better explanation

units should be written separately from the numbers